# Membrane Contact Sites in Autophagy

**DOI:** 10.3390/cells11233813

**Published:** 2022-11-28

**Authors:** Emma Zwilling, Fulvio Reggiori

**Affiliations:** 1Department of Biomedicine, Aarhus University, Ole Worms Allé 4, 8000C Aarhus, Denmark; 2Aarhus Institute of Advanced Studies (AIAS), Aarhus University, Høegh-Guldbergs Gade 6B, 8000C Aarhus, Denmark

**Keywords:** phagophore, autophagosome, endoplasmic reticulum, mitochondria, MAMs, plasma membrane, lipid droplets, vacuole, lipid transfer

## Abstract

Eukaryotes utilize different communication strategies to coordinate processes between different cellular compartments either indirectly, through vesicular transport, or directly, via membrane contact sites (MCSs). MCSs have been implicated in lipid metabolism, calcium signaling and the regulation of organelle biogenesis in various cell types. Several studies have shown that MCSs play a crucial role in the regulation of macroautophagy, an intracellular catabolic transport route that is characterized by the delivery of cargoes (proteins, protein complexes or aggregates, organelles and pathogens) to yeast and plant vacuoles or mammalian lysosomes, for their degradation and recycling into basic metabolites. Macroautophagy is characterized by the de novo formation of double-membrane vesicles called autophagosomes, and their biogenesis requires an enormous amount of lipids. MCSs appear to have a central role in this supply, as well as in the organization of the autophagy-related (ATG) machinery. In this review, we will summarize the evidence for the participation of specific MCSs in autophagosome formation, with a focus on the budding yeast and mammalian systems.

## 1. Introduction

### 1.1. Membrane Contact Sites

Membrane contact sites (MCSs) are formed by stable association between regions of the limiting membrane of two or more organelles, which are physically tethered together and exert specific functions [1,2]. MCSs are very dynamic; they can rapidly assemble and disassemble, according to their function [3,4], and the gap between the adjacent membranes is highly variable. In yeast, for instance, some describe the minimal distances between the ER and plasma membrane (PM) as 12−13 nm [5], whereas others state 16 nm and a maximal distance up to 59 nm [6]. In COS-7 cells, the membrane distance at the ER-PM MCSs has been reported to be on average 18.8 nm [7], while it is between 10−25 nm in Jurkat T-cells [8]. Fusion of the opposing membranes at the MCSs is prevented through the repulsion forces generated by the negatively charged phosphate head groups of the phospholipids in the membranes and/or potentially by the proteins residing in the MCSs, which could be acting as spacers [1]. It is also likely that the number of proteins present at the MCSs already serve as a barrier, as well as the absence of fusogenic factors, such as the SNARE proteins [1].

MCSs are ubiquitous and are often established between multiple organelles [9]. Therefore, rather than looking at organelles as separated entities, they must be seen as components of a vast spatiotemporally regulated network, composed of all the intracellular compartments and the PM, which exert their functions in an orchestrated and coordinated manner. Metabolic channeling is a key function of MCSs. The process has been implicated in calcium exchange, and MCSs of the ER—along with other organelles, such as mitochondria, PM and Golgi—are involved in lipid biosynthesis and exchange, and consequently, also organelle biosynthesis [2,10]. Lipid biosynthesis and exchange via MCSs also play a pivotal role during macroautophagy.

Multiple proteins can be part of a single MCS, and depending on their functional contribution, they are subclassified into tethers, functional proteins or regulatory proteins (Figure 1) [1].

Tethers are typically composed of two proteins, one each on the opposing organelles; through their binding, they establish and maintain the physical connection between the adjacent membranes, thus generating the MCSs. Examples are Nvj1 and Vac8 at the yeast nucleus-vacuole junction, or tricalbins and extended synaptotagmins (E-Syts) at the yeast and mammalian ER-PM MCSs [11,12]. Multiple tether pairs can play a role in the generation of the same MCSs, e.g., in yeast, ER-PM MCSs involve at least six tether pairs, namely the three tricalbins localized at the PM (Tcb1, Tcb2 and Tcb3) and three integral ER membrane proteins (Scs2, Scs22 and Ist2) [12,13]. Determining complete tether sets per MCS is challenging as the concomitant deletion of the aforementioned six tether pairs fails to dissolve all the ER-PM MCSs [13], highlighting the existence of MCSs between the same organelles involving other sets of tethers. Functional proteins exert MCS-specific functions such as lipid transfer, e.g., the STARD3 protein, which transfers cholesterol from the ER to endosomes at the endosome-ER MCSs [14]. Interestingly, STARD3 also acts concomitantly as a tether for endosome-ER MCSs together with its ER binding partners, the VAMP-associated proteins (VAPs) VAPA and VAPB. Finally, regulatory proteins recruit functional and/or tether proteins to the MCSs, can redistribute proteins within the MCSs to alter membrane properties, and activate or inactivate the functional proteins [1]. Thus, the regulatory proteins also modulate the function of the MCS. For example, yeast Osh3, which localizes at ER-PM MCSs, acts as a phosphatidylinositol-4-phosphate sensor at the PM to activate downstream actors in phosphatidylinositol (PtdIns) biosynthesis in the ER [15]. Another example is RAB32, which localizes to specific mitochondria-associated ER membranes (MAMs) and mediates the recruitment of both protein kinase A (PKA) and the chaperone protein calnexin via its GTPase activity [16]. While accumulation of calnexin alters calcium release from the ER, PKA redistribution increases the phosphorylation status of downstream targets, such as BAD. The combined effect of PKA and calnexin enhances apoptosis.

### 1.2. The Mechanism of Macroautophagy

The term autophagy entails all those transport pathways that deliver intracellular components to vacuoles/lysosomes for turnover. Currently, three main autophagic processes have been described: (1) macroautophagy, which is characterized by the sequestration of the cargoes by double-membrane autophagosomes that fuse with vacuoles or lysosomes; (2) microautophagy, which involves the direct engulfment of the cargo by the endosomes or vacuoles/lysosomes [17]; and (3) chaperone-mediated autophagy (CMA), in which single polypeptides with a specific recognition sequence are recognized by HSP70 and translocated into the mammalian lysosomes via a channel formed by LAMP2A [18].

Macroautophagy, hereafter autophagy, is the most studied type of autophagy, and it is highly conserved amongst eukaryotes. Autophagosomes are formed de novo through a mechanism that is characterized by five steps: (1) initiation and phagophore nucleation, (2) phagophore elongation, (3) phagophore closure, (4) autophagosome maturation and fusion with the vacuole/lysosome, and (5) degradation of the inner autophagosomal membrane and the cargo (Figure 2A). Autophagosome biogenesis is mediated by the so-called autophagy-related (ATG) proteins, and the core machinery has been sub-grouped into five functional modules: the Atg1/ULK kinase complex, the Atg9/ATG9A-containing vesicles, the autophagy-specific phosphoinositide 3-kinase (PI3K) complex, the Atg2/ATG2-Atg18/WIPI4 lipid transfer complex and two ubiquitin-like conjugation systems [19,20].

Autophagy is induced by different cellular and environmental cues (Figure 2A) [21]. For instance, autophagy is triggered upon nutrient starvation, and through the bulk degradation of cytoplasm and organelles, it maintains the homeostasis of metabolites, such as lipids and amino acids [22]. Two well-conserved key regulators of autophagy are mechanistic target of rapamycin kinase complex 1 (mTORC1) and AMP-activated kinase (AMPK), which through phosphorylation, negatively and positively regulate autophagy induction in response to amino acids and ATP levels, respectively [23,24,25,26]. Autophagosomes are generated at specific subcellular locations called phagophore assembly site or pre-autophagosomal site (PAS) in yeast or omegasomes in mammalian cells (Figure 2B,C) [27,28,29,30,31]. The PAS is formed in between the vacuole and the ER, while the omegasomes are a subdomain of the ER enriched in phosphatidylinositol-3-phosphate (PtdIns3P) (Figure 2B,C) [29,32,33]. The concerted action of the Atg1/ULK kinase complex, Atg9/ATG9A-containing vesicles and the PI3K complex at these sites leads to the nucleation of a membranous cistern known as the phagophore or isolation membrane (Figure 2A) [19,20,34,35,36,37]. The seeding membranes for the phagophore nucleation are not well characterized, and, in addition to the Atg9/ATG9A-containing vesicles, it is likely that several membrane sources contribute, including vesicles derived from the ER, the ER-to-Golgi intermediate (ERGIC) and/or the recycling endosomes [35,36,37,38,39,40]. The PtdIns3P synthesized at the phagophore by the PI3K complex is key for the subsequent recruitment of the rest of the ATG machinery. In particular, this phospholipid triggers the association of effectors like the ones belonging to the WD-repeat protein interacting with the phosphoinositides (WIPI) protein family [19,20,34]. Atg21/WIPI2 and Atg18/WIPI4 mediate the association of the two ubiquitin-like conjugation systems and the assembly of the Atg2-Atg18/ATG2-WIPI4 lipid transfer complex at the phagophore, respectively [19,20,34]. The engagement of these functional modules is essential for the phagophore expansion and phagophore closure (Figure 2A), which appears to involve the ESCRT system [41,42].

Complete autophagosomes undergo a maturation step that is characterized by the hydrolysis of PtdIns3P into PtdIns by Ymr1 in yeast, MTM-3 in *Caenorhabditis elegans* and MTMR8 in mammals—all members of the myotubularin phosphatase family—and the release of the ATG proteins from their surface [19,20,34,43,44,45]. While in yeast they fuse directly with the vacuolar membrane (Figure 2A), autophagosomes are transported via microtubules to the microtubule-organizing centers (MTOCs), where late endosomes and lysosomes are concentrated [42,46]. These autophagosomes fuse first with late endosomes and then lysosomes, or directly with lysosomes (Figure 2A), to form amphisomes and autolysosomes [47]. Fusion is mediated by SNARE proteins and requires prior tethering, an event that involves the small GTPase RAB7, its guanosine exchange factor and the homotypic fusion and protein sorting (HOPS) complex [41,47,48]. Finally, upon fusion, the inner membrane of the autophagosomal vesicle and its cargo are degraded by vacuolar/lysosomal hydrolases and the resulting metabolites transported into the cytosol to be used as a source of energy or building blocks for the biosynthesis of macromolecules [22]. As it will be discussed in the subsequent section, membrane supply for the biogenesis and subsequent expansion of phagophores requires, amongst other factors, MCSs between the phagophore and multiple intracellular organelles, including ER, and possibly mitochondria and lipid droplets (LDs).

## 2. MCSs during Autophagosome Biogenesis

### 2.1. ER-Phagophore MCSs

The ER plays a key role in the lipid biosynthesis as it serves as a central cellular hub for intracellular communication, organelle trafficking and macromolecule synthesis. Therefore, it forms MCSs with all intracellular organelles, including the phagophore whose extremities are tethered with the ER in both yeast and mammalian cells [31,32,33,49,50,51,52,53]. In mammals, it appears that the phagophores also have some MCSs with the ER distributed elsewhere on their surface [54,55]. While these latter MCSs remain to be characterized, work in yeast has identified some of the components and the mechanism establishing the MCSs between the extremities of the phagophore and the ER (Figure 3A). This association seems to be less strong than the one with other contacts (e.g., the nucleus) since the level of visible deformation on both organelles is much lower [31]. The fact that the phagophore is very frequently tethered via part of its rim to the ER or the nucleus is consistent with ongoing lipid transfer at these MCSs [31,56]. In those contacts, Atg9-Atg2-Atg18 complex is positioned at the phagophore tips, while ER exit sites (ERES) are present on the ER side (Figure 3A) [32,33]. ER-phagophore MCSs are probably tethered by Atg2-Atg18 and ATG2A-WIPI4 [57,58]. Structural studies suggest in vitro that ATG2A, which has a rod-like structure shared by all ATG2 proteins [58], tethers a liposome by binding with one of its edges the membranes containing Ptdlns3P and WIPI4, the mammalian Atg18 counterpart [19,20,34], and with the other opposite membrane [59,60,61].

In vivo, Atg2 is recruited to the phagophore via coincidence binding to PtdIns3P and Atg9, which concentrates on high curvature membrane regions such as the extremities of the phagophore [52]. Atg2 binding (via its C-terminal region) to Atg9 induces the subsequent recruitment of Atg18, which bind to both Atg2 and PtdIns3P (Figure 3A) [52,62,63]. The N-terminal domain of Atg2, in contrast, appears to be responsible for its interaction with the ER [57], but it remains enigmatic which ERES components bind to ATG2 proteins. In addition to acting as a tether, in vitro experiments have revealed that ATG2 proteins act as functional proteins as they are able to transfer lipids at the phagophore-ER MCSs [58,61,64]. Together with the VPS13 proteins, which associate with several MCSs [65,66], ATG2 proteins belong to the chorein_N family of lipid transporters. The rod-like structure of ATG2 proteins possess a N-terminally chorein_N motif that allows the binding and extraction of multiple glycerolipids from the donor membrane in vitro [58,61,67]. The subsequent phospholipid transfer occurs through a hydrophobic cavity that spans their entire length and can accommodate several glycerolipids [61,67]. Atg18 and WIPI4 binding accelerate the lipid transfer by ATG2 proteins, indicating that those proteins may act as MCSs regulators [60,64]. The lipid transfer by ATG2 proteins could lead to an amassment of phospholipids on the external lipid layer of the phagophore membrane, disrupting its expansion into an autophagosome. This potential problem, however, is very likely avoided by Atg9/ATG9A proteins, which, beside their role in phagophore nucleation, also have intrinsic lipid scramblase activity shown in in vitro experiments [68,69] and directly interact with at least two regions of ATG2 proteins [52,62,63]. Consistent with this notion, Atg9/ATG9A lipid scramblase mutants show a defect in phagophore expansion [68,69], and recent in vitro experiments have revealed that Atg9 stimulates the Atg2-Atg18 complex-mediated lipid transfer to the acceptor membrane [70]. Moreover, evidence suggests that ATG9-ATG13-ATG101 is central to the formation of a super-complex with the ULK kinase and PI3K complexes, which enhances the action of the ATG2-WIPI4 complex as a tether and lipid transfer protein at the phagophore-ER MCSs [71].

Extraction of lipids from the external leaflet of the ER could also create an unbalanced lipid distribution in the limiting membrane of this organelle, which would need to be dissipated to guarantee an uninterrupted flux of lipids. Interestingly, two redundant ER-localized mammalian lipid scramblases, i.e., VMP1 and TMEM41B, are essential for autophagy [56,72,73,74]. In line with this idea, it has been observed that Atg9, here acting as regulatory protein, also enhances Atg2 lipid transfer when localized at the donor membrane [70]. 

The VMP1 scramblase—which localizes to specific mitochondria-ER MCSs, ER-lipid droplets and ER-endosome MCSs—plays an important role in mammalian autophagy because when depleted, it leads to an autophagic flux block and concomitant high levels of PtdIns3P (see also below) [75,76]. VMP1 activity appears to be a MCS regulator that ensures the appropriate size of the MAMs, since its depletion results in abnormal MAM phenotypes [73]. VMP1 also directly modulates ER-phagophore MCSs, preventing SERCA-pump inactivation by blocking binding of phospholamban (PLN)/sarcolipin (SLN), two micropeptide proteins, to SERCA (Figure 3A) [48]. Dysregulated SERCA activity disrupts calcium homeostasis and leads to an induction of autophagy [77,78]. VMP1 depletion, on the other hand, results in the failure of the phagophore to mature, which is accompanied by an increased interaction at omegasomes between the VAPs, acting here as regulatory proteins, with the autophagy proteins FIP200, a subunit of the ULK kinase complex, and WIPI2, a component of the ubiquitin-like conjugation systems (Figure 3A) [48,79]. VAPs are involved in the establishment of several MCSs between the ER and other organelles, including the phagophore [49,80]. Consequently, depletion of VAPs also results in impaired autophagy because, together with WIPI2, they tether phagophores with the ER (Figure 3A) [79]. Interestingly, mutations in VAPB are associated with amyloid lateral sclerosis, and one of them, VAPB^P56S^, leads to a reduced autophagic flux [79]. However, it cannot currently be excluded whether the autophagy defect of the VAPB^P56S^-expressing cells is indirectly due to a defect in the correct localization of the endolysosomal degradative enzymes caused by an alteration of the ER-Golgi MCSs [81]. VAP proteins participate together with the ER integral membrane proteins ATLASTIN 2 (ATL2) and 3 (ATL3) in recruiting the ULK kinase complex onto the ER by interacting with ULK1 and ATG13, and this event enhance MCSs formation [82].

### 2.2. ER-Mitochondria MCSs and Mitochondria-Associated ER Membranes (MAMs)

ER-mitochondria MCSs and MAMs are ubiquitous, with about 5 to 20% of the mitochondrial network linked to the ER, and even more under specific stress conditions, such as nutrient starvation or tunicamycin treatment [84,85,86]. Mitochondria have a long-standing history as potential suppliers for autophagosomal membranes [29,54,55,87,88]. Initially, it was proposed that mitochondria are the main membrane source for the lipids composing autophagosomes [87]. However, later it was shown that not mitochondria, but rather MAMs are involved in autophagosome biogenesis by promoting the recruitment of ATG14 to the ULK kinase complex already present at the phagophore nucleation sites [89]. ATG14 recruitment to the MAMs depends on the SNARE protein STX17, and it enhances the assembly of the autophagy-specific PI3K complex as well as its lipid kinase activity [89]. In line with this, depletion of STX17 through bacterial proteases results in decreased autophagy [90]. However, STX17 is also required for fusion between autophagosomes and lysosomes [91,92,93]. Consequently, the autophagy flux block observed upon inactivation of STX17 is probably due to an impairment of at least two autophagy steps. Furthermore, ER-mitochondria MCSs and MAMs may contribute to autophagosome biogenesis through their function in lipid metabolism and synthesis [1,3,94,95]. 

MAMs can be formed by several tether pairs, which are characterized by specific predominant functions (Figure 3B). The MAMs composed by the tether pair VAPB (in the ER) and PTPiP51 (in mitochondria) influence autophagy initiation through calcium signalling [96]. Dissolving these MCSs by ablating VAPB or PTPiP51 leads to an enhancement in autophagosome formation. These MAMs are relevant for calcium transfer from ER to mitochondria [80], and impairment of this function appears to lower mTORC1-mediated autophagy inhibition. The full connection between calcium signalling and autophagy needs further elucidation, although it has recently been shown that SERCA interacts with STX17, which is required for early and late autophagy steps [97].

The ER transmembrane proteins EI24 and IP3R, together with GRP75 and outer mitochondrial membrane protein VDAC1, form another set of MAMs (Figure 3B) [98]. Depletion of EI24 causes an impairment of autophagosome biogenesis and autophagic flux [83,99], suggesting that those MCSs also play a key role in autophagy.

Moreover, inhibition of oxidative phosphorylation (OXPHOS) or nutrient starvation results in enhanced MAMs formation with MFN2 as a tether (Figure 3B) [100]. Increased MAM formation is regulated by AMPK, which phosphorylates MFN2 upon the decrease of cellular energy levels. How this affects the cell remains unclear, since together with MFN1, MFN2 is a mitochondrial fusion factor [101]. MFN2 appears to have an impact on glycolysis and OXPHOS because energetic stress in MFN2 knockout cells leads to a reduced glycolysis and OXPHOS capacity. Interestingly, MFN2 forms a complex with ERLIN1, AMBRA1 and GD3 in MAMs-specific lipid microdomains (Figure 3B) [86]. These microdomains participate in the initial steps of autophagy in several ways [86,102,103]. ERLIN1, but also ERLIN2, specifically accumulate in MAMs-associated ER lipid microdomains during lowering of the cellular energy [86,104]. ERLIN1 interacts with AMBRA1 (Figure 3B) [102], which is a well-known regulator of autophagy that is phosphorylated by ULK1 during starvation and promotes autophagy by connecting the autophagy-specific PI3K complex to dynein for its transport to omegasomes [105,106]. The ganglioside GD3 is a component of the MAM’s lipid raft that interacts with PtdIns3P and is also present at sites of autophagosome formation [103]. Depletion of GD3-synthetase results in an impaired autophagic flux.

In yeast, MAMs have been implicated in basal mitophagy, but not in bulk autophagy [107]. In particular, it appears that the phagophore forms at the ER-mitochondria MCSs called ER mitochondria encounter structure (ERMES) and composed by Mmm1 and Mdm12 at the ER membrane and Mdm10 and Mdm34 at the mitochondrial membrane (Figure 3B) [107]. In mammals, MAMs that contain BECN1 and PTEN-induced kinase 1 (PINK1) are also required for ER-mitochondria MCS formation, both promoting membrane tethering and mitophagosome biogenesis, although it remains to be determined whether they represent the counterpart of the yeast ERMES [108]. The PINK1 kinase, together with E3 ubiquitin ligase PARKIN, modulates certain types of mitophagy, especially those induced by mitochondrial depolarization [109]. During the early steps of mitophagy, ER-mitochondria MCSs must be dissolved [110,111] and this seems to require the phospho-ubiquitination of their tether MFN2 by PINK1 and PARKIN (Figure 3B), which triggers p97-mediated MFN2 degradation by the proteasome [111,112,113]. This finding is in contrast with the above-mentioned one in which the conclusion was that mitophagy induction increases MAM formation in a PINK1-dependent way. The reason for this apparent discrepancy between studies is unknown. 

In conclusion, it appears that multiple different types of ER-mitochondria MCSs are important for the normal progression of autophagy. Their functional interconnection, however, remains a question to be addressed. Nonetheless, at the cytosol-ER interface, a very recent study seems to functionally connect EI24, FIP200 and calcium signalling [83], and, thus, also some of the MAMs/ER-mitochondria MCSs and phagophore-ER MCSs that we have described. 

### 2.3. ER-PM MCSs

The ER also forms MCSs with the PM, which are involved in lipid trafficking and calcium homeostasis, but autophagy regulation, as well [114,115]. ER-PM MCSs may contribute to one or more steps necessary for the generation of the phagophores [116]. In particular, the mammalian ER-PM MCSs composed by the tethers ESYT1 at the ER, and ESYT2 and ESYT3 at the PM, appear to have a central role in modulating autophagosome biogenesis; consequently, autophagy inducers enhance the expression of ESYT2, as well [116]. ESYT proteins are required for autophagosome formation at the ER-PM MCSs by locally promoting the interaction between VMP1 and BECN1, which leads to an enhancement of the PI3K complex activity [116]. While VMP1 and ESYT2 form a stable complex, the well-known interaction between VMP1 and BECN1 appears to be induced by stress conditions [116,117,118]. Since ESYTs colocalize with LC3 and other autophagy marker proteins, this evidence points to the ESYT protein-containing MCSs being a platform to enable PtdIns3P synthesis [116]. 

### 2.4. Phagophore-Vacuole MCSs 

The interaction of the yeast Atg machinery with the vacuole is not only important for the fusion of the mature autophagosomes with vacuoles, but also for the early steps of autophagy. The PAS is formed adjacently to the vacuole (Figure 2B) [27,30], and its formation involves the initial recruitment of the Atg1 kinase complex via binding to the vacuolar surface protein Vac8 via Atg13 (Figure 3C) [119,120,121,122,123]. The subsequent local generation of the phagophore probably leads to the establishment of the so-called vacuole-isolation membrane contact sites (VICS). This single MCS is present over the course of the entire phagophore elongation [32,33]. Electron tomographic analyses have revealed that the phagophore is associated with the vacuole with the side or the back, such that the elongation occurs toward the vacuolar membrane [31]. The distance and area of contact between the vacuole and the phagophore is very variable, indicating that this MCS is not specifically structured through for instance spacers [31]. In the absence of Vac8, the PAS is no longer associated with the vacuole, and autophagosome biogenesis only takes place proximal to the ER [120]. The resulting autophagosomes are smaller and formed at lower frequency, leading to an overall reduced autophagic flux [119,120,123,124]. Vac8 is also important for the progression of different types of selective autophagy, including the cytoplasm to vacuole targeting (Cvt) pathway, mitophagy, pexophagy and ribophagy [122,125,126]. In the context of selective types of autophagy, it has been postulated that Vac8 organises the PAS by binding to Atg11, which allows both the recruitment of the cargo near the vacuole and Atg1 kinase complex activation, thereby promoting the sequestration of the targeted cargo into the nascent autophagosome [127]. Vac8 also interacts with its armadillo repeat (ARM) domains within the Vps15-Vps34 subcomplex that is part of the autophagy-specific PI3K complex (Figure 3C) [125,127]. This interaction is important for recruiting the autophagy-specific PI3K complex to the PAS for the synthesis of Ptdlns3P and the mobilization of downstream PtsIns3P-binding Atg factors [125,127]. It still remains to be determined when exactly the VICSs are formed, i.e., with the vesicles initially at the PAS or with the phagophore nucleated at this location. Nonetheless, VICS are also characterized by the presence of Atg21 and the absence of the vacuolar membrane protein Vph1 [123]. Atg21 is recruited to the phagophore via binding to PtdIns3P and is required to guide Atg8 lipidation [19,20,34,123]. The mechanism leading to the preferential concentration of Atg21 at the VICS is unknown. ARMC3, a mammalian ortholog of Vac8 with spatiotemporal expression in testis tissues, is implicated in ribophagy during spermatogenesis, but its involvement in MCSs remains to be elucidated [125,126].

### 2.5. Nucleus-Vacuole Junction MCSs

Nucleus-vacuole junctions (NVJs) are an extended MCSs between the yeast vacuole and its nucleus (Figure 3D) [128]. They play a central role in nucleophagy, the selective degradation of nuclear material, which includes RNA, nuclear envelope and lamina and/or parts of the spindle apparatus by autophagy [129]. Two mechanisms of nucleophagy have been described in yeast; selective microautophagy, or piecemeal nucleophagy (PMN), and macroautophagy [130]. Both mechanisms result in the destruction of non-required nuclear parts [131,132]. NVJs are required for PMN, and their inward invagination into the vacuole followed by pinching off leads to the degradation of small parts of the nucleus, i.e., micronuclei [131]. The NVJ tethers are the nuclear membrane protein Nvj1 and vacuolar Vac8 (Figure 3D) [133,134,135]. Nvj1 is also degraded during PMN [130]. Although NVJs are constitutively formed, NVJ1 is equipped with a promoter with two stress response elements (STRE), and upon stress-inducing conditions, such as nutrient deprivation or hypoxia, the elevated amounts of Nvj1 correlates with an enhanced number of NVJs [131,132,133]. The binding domain between Vac8 and Nvj1 seems to be similar to the one mediating the Vac8-Atg13 association [136]; how the two interactions are reciprocally regulated remains to be determined. 

## 3. Conclusions

As highlighted in this review, several MCSs play an important role in autophagy regulation and progression (Figure 4). However, the architecture of most MCSs has been poorly defined, and the mechanism by which they are assembled and function remains to fully determined. Moreover, it has to be established whether some MCSs only operate under specific growing and/or metabolic conditions and what is their specific role. For instance, several MCSs, including the ones between the phagophore and either the ER or PM, have been proposed to be lipid donors during autophagosome biogenesis, but it is unclear why lipids are derived from different sources. Potentially, autophagosome biogenesis could utilize different MCSs depending on the conditions triggering this pathway, obtaining lipids from the most optimal source. Within the same organism, cells composing the different tissue have relatively different organelle composition and dimensions. Thus, it also may be possible that specific MCSs contribute to autophagy in a cell-type specific manner, even under the same autophagy-inducing conditions. On this line, it also remains the be explored whether the protein composition and function of determined MCSs may vary depending on the cues triggering autophagy.

Although MCSs have been a cell biology area of intense investigation during the last decades, the ones involved in autophagosome biogenesis are still marginally understood. Future studies will help to decipher their organization, regulation and function, which may also provide new links to other cellular functions intimately connected with autophagy, such as metabolism and immunity, for example.

## Figures and Tables

**Figure 1 cells-11-03813-f001:**
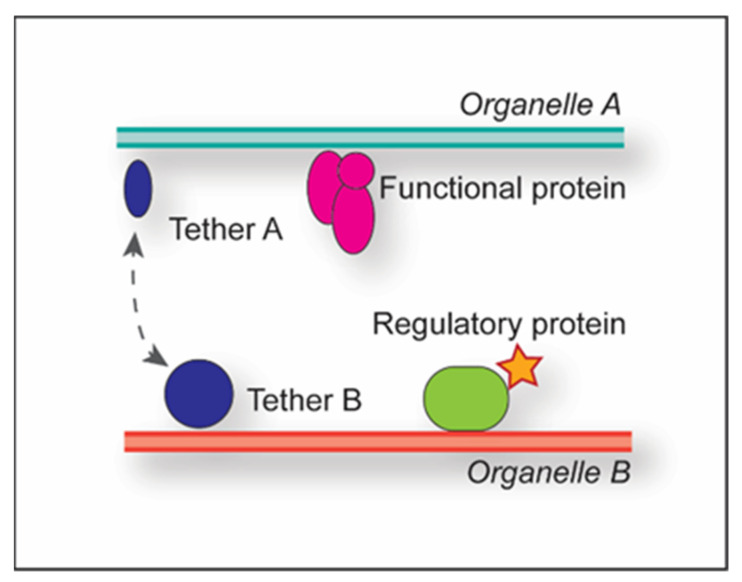
Classification of proteins in MCSs. MCS-associated proteins can be classified into tether pairs present on both organelles (indicated by A and B) (blue), functional proteins (pink) and regulatory proteins (green). The dashed grey line highlights interaction. The red star indicates activation.

**Figure 2 cells-11-03813-f002:**
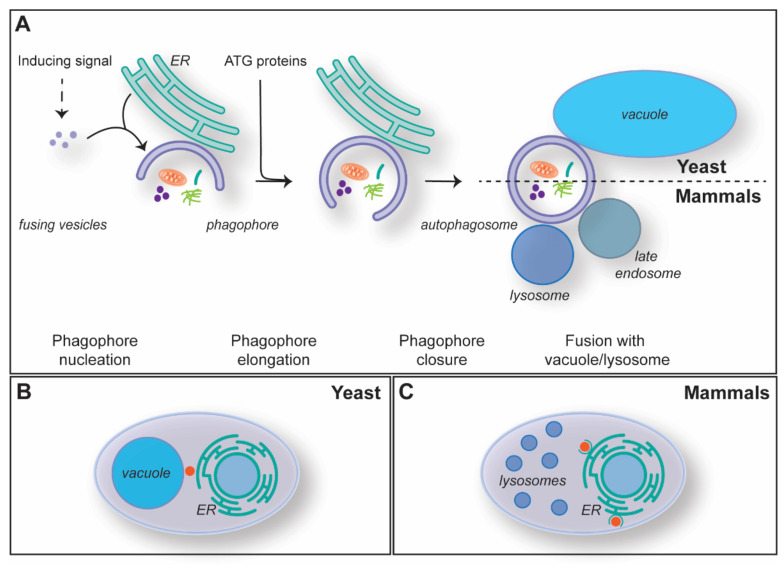
The mechanism of macroautophagy. (**A**) The mechanism of autophagosome formation and its key steps. (**B**) Subcellular distribution of the PAS (red dot) in the yeast Saccharomyces cerevisiae, which is localized in between the ER and the vacuole. (**C**) The sites of autophagosome formation (red dot) in mammalian cells are adjacent to the ER in specialized subdomains known as omegasomes.

**Figure 3 cells-11-03813-f003:**
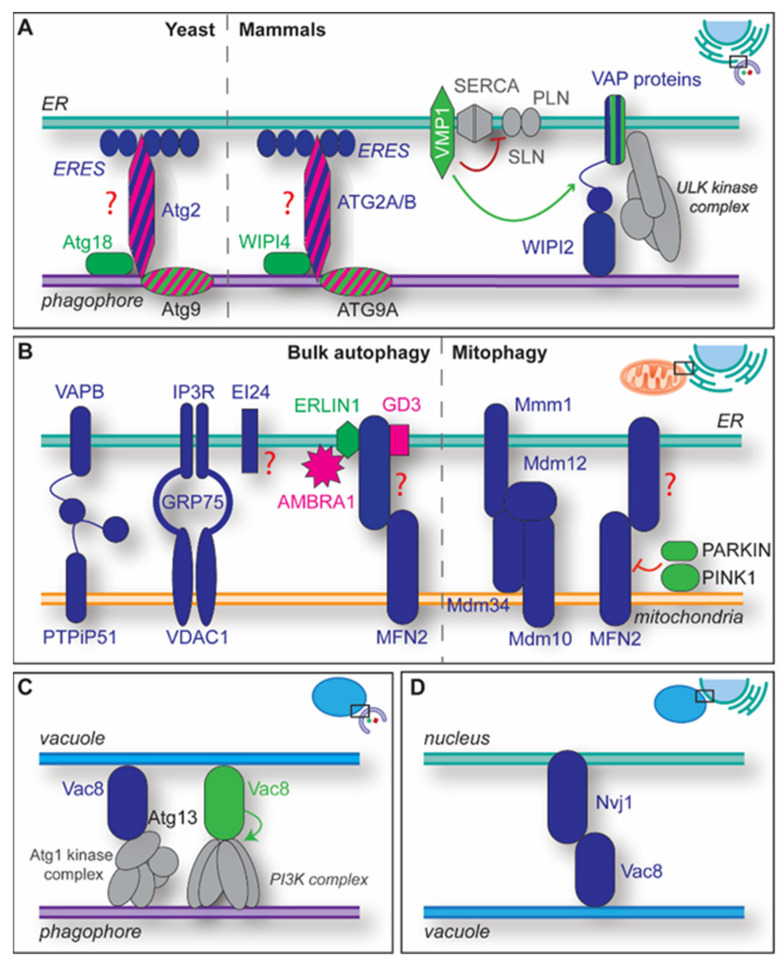
Overview of some of the MCSs playing a relevant role in autophagy. (**A**) One type of ER-phagophore MCSs involve Atg2/ATG2A/ATG2B, Atg9/ATG9A, Atg18/WIPI4 and the ERES. The Atg2/ATG2A/ATG2B tether partner at the ERES remains to be identified, which is indicated by the red question mark. Atg2/ATG2A/ATG2B very likely supply the phagophore with lipids. In addition to probably assisting the transfer of lipids by ATG2 proteins, the VMP1 lipid scramblase exerts a regulatory role at the phagophore-ER MCSs by preventing the inactivation of the ER-localized calcium pump SERCA (red line) by PLN/SLN binding (red arrow). VMP1 also appear to be a regulator (green arrow) of MCSs between the phagophore and the ER by modulating the interaction between VAP proteins and both FIP200, which stabilizes the ULK kinase complex (grey) at the omegasome, and WIPI2. (**B**) ER-mitochondria MCSs and MAMs are central for autophagosome biogenesis and several of them have been described. Their interrelationship remains largely unknown, i.e., it is unclear if the different MAMs implicated in autophagy are part of the same MCSs. VAPB and PTPIP51 form one of the MAM tethers, the complex of VDAC1, IP3R and GRP75 another one. EI24 interacts with the subunits of this complex, indicated by the grey dashed lines, and is required for normal autophagic flux. However, the role of EI24 in autophagy is mechanistically not well understood, represented by the red question mark, although a very recent study connects it to calcium signaling [83]. The tether MFN2 forms complexes with GD3, its binding partner, ERLIN1, and AMBRA1 in MAM-specific lipid-microdomains, to regulate autophagosome biogenesis. ERLIN1 and GD3 are associated to lipid-rafts in the ER, and ERLIN1 and AMBRA1 interact in those. AMBRA1 is phosphorylated by ULK1 upon autophagy induction and recruits the autophagy-specific PI3K complex to the omegasome. In yeast, the ERMES formed by Mdm10, Mdm12, Mdm34 and Mmm1 are key for mitophagy progression. In mammals, during PINK1-dependent mitophagy, MFN2 tethers are disassembled through the action of PINK1 and PARKIN, indicated by the green dashed arrow, which leads to the proteasomal degradation of MFN2. (**C**) The vacuole-phagophore MCSs, which are involved in non-selective and selective types of autophagy in yeast, are formed by interaction of Vac8 with the Atg1 kinase complex, in particular with the Atg13 subunit. In those MCSs, Vac8 also recruits concomitantly the autophagy-specific PI3K complex via interaction with its subcomplex Vps15-Vps34 (**D**) The nucleus-vacuole MCSs are involved in yeast PMN and are generated by an interaction between Nvj1 and Vac8. Proteins colored in blue are tethers, proteins colored in green have a regulatory function, while the one in pink represents functional proteins. ATG complexes are in grey. Proteins striped in dual colors exert the two functions displayed by each color.

**Figure 4 cells-11-03813-f004:**
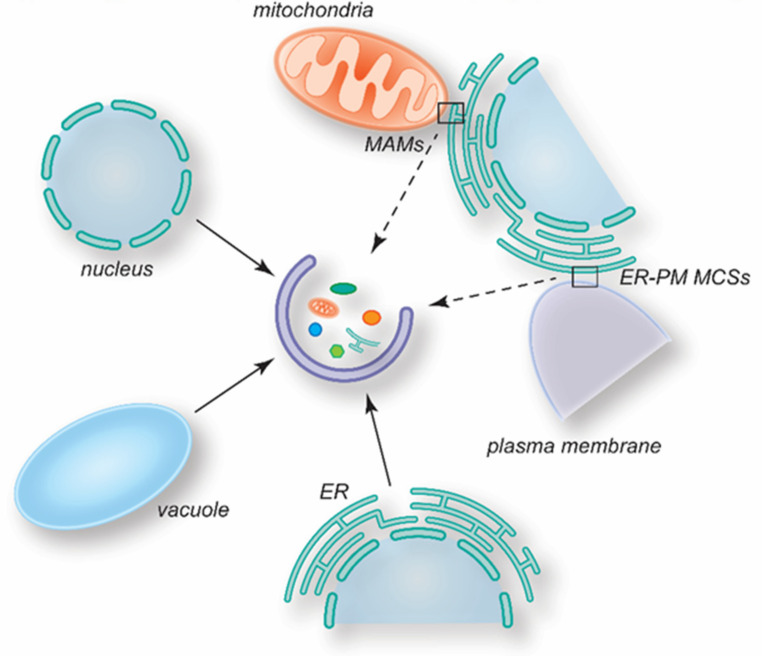
Overview of the MCSs involved in autophagosome biogenesis. Several MCSs play a role in the regulation of autophagy, contributing in different ways. Continuous arrows highlight MCSs between organelles and the phagophore. Dashed arrows indicate MCSs between organelles that participate in the formation of autophagosomes.

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
