# Peer review of "Membrane Contact Sites in Autophagy"

_cells, 2022, doi:10.3390/cells11233813_

Round 1
Reviewer 1 Report
Zwilling and Reggiori summarize the role of membrane contact sites in autophagy well. I find the manuscript suitable for publication as I only have minor comments, listed below:
1) Abstract starts odd. It would benefit from rewriting, especially the first sentence which is partly incorrect. The communications between organelles do not only occur “via phospholipid transfer in [at] membrane contact sites”.
2) Line 32: “(REF)”. references missing.
3) Starting from Line 53-54: The difference between functional and regulatory proteins is not clear. All proteins are somehow functional proteins, including regulatory and tether proteins have functions too. Authors also mention that regulatory proteins have “regulatory function”, thus making them also functional proteins. This needs to be sorted out.
4) Starting from Line 77: Please note that mitochondria-associated membranes (MAMs) are not the same as ER-mitochondria contact sites. It is true that many elements of MAMs and contact sites overlap, but still not the same thing. Please correct that.
5) Line 213 mentions “VAPs” but this is already defined earlier in the text as VAPA and VAPB.
6) Line 225: “an d” should be “and”
7) Lines 257-260 should be part of the figure legend.
8) Lines 262-263: “MAMs are ubiquitous, with about 5 up to 20% of shared surface between [the] ER and mitochondria, and even more under non-physiological conditions”. 5-20% of the surface of what? Is up to 20 of mitochondria surface is shared with the ER? Or the other way around. Do authors mean up to 20% of the shared surface between the ER and mitochondria? In that case, what is the remaining 80% of the shared surface between the organelles? They would be also contact sites.
Also, the term “non-physiological conditions” is a bit vague.
9) Line 278: “PTPiP51” should be PTPIP51.
10) The color coding in figure 3 is extremely confusing. What is “gray/black”? Dual coloring is also confusing; looks as it is a protein complex (also see point 3 above). What are the proteins colored in navy/dark blue?
Author Response
We thank the two reviewers for their overall very positive evaluation of our review and their constructive feedback, which we believe that has helped to improve our manuscript. A point-by-point response to each comment is given below, and the related changes and corrections are highlighted in blue in the manuscript.
1) Abstract starts odd. It would benefit from rewriting, especially the first sentence which is partly incorrect. The communications between organelles do not only occur “via phospholipid transfer in [at] membrane contact sites”.
Thank you for this comment. We have deleted phospholipid transfer from the first sentence and changed the two following ones to generalize the functions of the MCSs. As a result, the first three sentences of the abstracts have been modified from ¨Eukaryotes utilize different communication strategies to coordinate processes between different cellular compartments either indirectly, through vesicular transport or directly, via phospholipid transfer in membrane contact sites (MCSs). For example, the endoplasmic reticulum (ER)-mitochondria and the ER-plasma membrane MCSs have been implicated in lipid metabolism, calcium signaling and the regulation of organelle biogenesis in various cell types. In addition, several studies have shown…¨ into ¨Eukaryotes utilize different communication strategies to coordinate processes between different cellular compartments either indirectly, through vesicular transport or directly, via membrane contact sites (MCSs). MCSs have been implicated in lipid metabolism, calcium signaling and the regulation of organelle biogenesis in various cell types. Several studies have shown…¨
2) Line 32: “(REF)”. references missing.
Done.
3) Starting from Line 53-54: The difference between functional and regulatory proteins is not clear. All proteins are somehow functional proteins, including regulatory and tether proteins have functions too. Authors also mention that regulatory proteins have “regulatory function”, thus making them also functional proteins. This needs to be sorted out.
Thank you very much for this comment, which we agree with. To clarify this aspect, we have added the sentence at line 81: ‘Thus, the regulatory proteins also modulate the function of the MCSs.’.
4) Starting from Line 77: Please note that mitochondria-associated membranes (MAMs) are not the same as ER-mitochondria contact sites. It is true that many elements of MAMs and contact sites overlap, but still not the same thing. Please correct that.
Thank you. We should say that as us, a lot of researchers use interchangeability the terms MAMs and ER-mitochondria MCSs. We have been unable to find in the literature a clear definition of these two terms, which could allow us to correctly distinguish the ER-mitochondria contacts described in our review. Nonetheless, we have adapted the manuscript as following to be consistent with the terminology used in each cited study:
Line 85: from ‘(…) which localizes to specific mitochondria- ER MCSs (…)’ to ‘‘(…) which localizes to specific mitochondria-associated ER membranes (MAMs) (…)’.
Line 214: from ‘(…) which localizes to specific MAMs, (…)’ to ‘(…) which localizes to specific mitochondria-ER MCSs, (…)’.
Line 271: from ‘(B) MAMs are central for (…)’ to ‘(…) (B) ER-mitochondria MCSs and MAMs are central for’.
Line 294: from ‘ER-mitochondria MCSs’ to ‘ER-mitochondria MCSs and mitochondria-associated ER membranes (MAMs)’.
Line 295: from ‘MAMs are ubiquitous, with about 5 up to 20% (…)’ to ‘ER-mitochondria MCSs and MAMs are ubiquitous, with about 5 up to 20% (…)’.
Line 309-310: from ‘Furthermore, MAMs may contribute to (…)’ to ‘Furthermore, ER-mitochondria MCSs and MAMs may contribute to (…)’.
Line 315: from ‘These MAMs are relevant for calcium transfer (…)’ to ‘These MCSs are relevant for calcium transfer (…)’.
Line 323: from ‘(…) suggesting that those MAMs also play a key role (…)’ to ‘(…) suggesting that those MCSs also play a key role (…)’.
Line 333: from ‘(…) accumulate in MAMs-associated lipid microdomains (…)’ to ‘(…) accumulate in MAMs-associated ER lipid microdomains (…)’.
Line 342: from ‘(…) phagophore forms at the MAMs (…)’ to ‘(…) phagophore forms at the ER-mitochondria MCSs (…)’.
Line 346: from ‘(…) are also required for MAM formation both promoting (…) to ‘(…) are also required for ER-mitochondria MCS formation both promoting (…)’.
Line 350: from ‘(…) MAMs must be dissolved (…)’ to ‘(…), ER-mitochondria MCSs must be dissolved (…)’.
Line 357: ‘(…) that multiple different types of MAMs are important for the normal progression of autophagy.’ To ‘(…) that multiple different types of ER-mitochondria MCSs are important for the normal progression of autophagy.’
Line 361: ‘(…) thus also some of the MAMs and phagophore-ER MCSs that we have described’ to ‘(…) thus also some of the MAMs/ER-mitochondria MCSs and phagophore-ER MCSs that we have described.’
5) Line 213 mentions “VAPs” but this is already defined earlier in the text as VAPA and VAPB.
We have now defined the term VAPs, i.e., VAMP-associated proteins (VAPs), when we cite the first time VAPA and VAPB (line 78).
6) Line 225: “an d” should be “and”
Corrected.
7) Lines 257-260 should be part of the figure legend.
Indeed, corrected.
8) Lines 262-263: “MAMs are ubiquitous, with about 5 up to 20% of shared surface between [the] ER and mitochondria, and even more under non-physiological conditions”. 5-20% of the surface of what? Is up to 20 of mitochondria surface is shared with the ER? Or the other way around. Do authors mean up to 20% of the shared surface between the ER and mitochondria? In that case, what is the remaining 80% of the shared surface between the organelles? They would be also contact sites.
Also, the term “non-physiological conditions” is a bit vague.
We have adapted the sentence to clarify in line 295: ‘MAMs are ubiquitous, with about 5 up to 20% of shared surface between the ER and mitochondria, and even more under non-physiological conditions’ into ‘MAMs are ubiquitous, with about 5 up to 20% of the mitochondrial network linked to the ER, and even more under specific stress conditions, such as nutrients starvation or tunicamycin treatment’.
9) Line 278: “PTPiP51” should be PTPIP51.
Corrected.
10) The color coding in figure 3 is extremely confusing. What is “gray/black”? Dual coloring is also confusing; looks as it is a protein complex (also see point 3 above). What are the proteins colored in navy/dark blue?
Thank you very much to point this out. We have modified the figure by changing the color of the complexes that are recruited to the MCS from dark grey/black to light grey, and deleted the indication of the subunit which is interaction partner of proteins forming the MCSs. These subunits are now only mentioned in the figure legend. We also changed the dual coloring of proteins with two roles to stripes in dual color. Accordingly, we have corrected the figure legend at line 271, from ‘(…) the ULK kinase complex at the omegasome (…)’ to ‘(…) stabilizes the ULK kinase complex (grey) at the omegasome (…)’, and at line 292, from ‘ATG complexes are in grey/black, with the subunit interacting with tethers in black. Dual colored proteins exert the two functions displayed by each color.’ to ‘ATG complexes are in grey. Proteins striped in dual colors exert the two functions displayed by each color.’ Furthermore, we have changed the color of all tether proteins to the same blue to avoid confusion.
Reviewer 2 Report
The review by Zwilling and Reggiori focuses on a very important and timely aspect of autophagy research and cell biology in general. In particular, it summarizes and discusses the current knowledge of how membrane contact sites and lipid transfer at these contacts drive and regulate autophagy as well as autophagosome biogenesis. In the past years fascinating discoveries with contributions from the authors’ lab have been made in this area and the review will be a very useful resource for a wider community.
The review is well-written, balanced and comprehensive. I have only a few points the authors should address.
1. Figure 1 is too simple in order to be useful. For example, it is not clear what tether A and B means and why an arrow points between them.
2. Figure 2A: It is not clear if the fusion of vesicles is a major driving force for phagophore expansion and the actual text focuses on lipid transfer. The figure should be modified in this respect.
3. Lines 120-121: the following references should be added as they suggest that ATG9 vesicles are important contributors to the membrane seeds (10.1126/science.aaz7714, DOI: 10.1101/2022.08.16.504143, DOI: 10.1101/2022.08.03.502680).
4. This paper, which appeared after the submission of this review should be added to the paragraph (starting at line 170) discussing the interaction between ATG2 and ATG9 (10.1016/j.molcel.2022.10.017).
5. Line 257-260: This part of the text seems to belong to the legend rather than the main text. The current formatting suggests otherwise.
6. One aspect the authors may want to briefly touch on in their discussion of MAMs (lines 261 – 325) is that they may function in lipid metabolism and synthesis and may therefore regulate autophagy and autophagosome biogenesis in this respect.
- Line 32: and actual reference should be inserted into (REF)
Author Response
We thank the two reviewers for their overall very positive evaluation of our review and their constructive feedback, which we believe that has helped to improve our manuscript. A point-by-point response to each comment is given below, and the related changes and corrections are highlighted in blue in the manuscript.
